# Student behavior at university: The development and validation of a 10-dimensional scale

**Natalia Maloshonok** *, **Kseniia Vilkova**

Center for Sociology of Higher Education, HSE University, Moscow, Russia

* nmaloshonok@hse.ru

## Abstract

This study proposes the multidimensional concept of 'student behavior at university' and methodology for its quantitative investigation. Unlike previous concepts related to aspects of the student experience, the idea of student behavior considers the combinations and interrelations of individual and environmental characteristics affecting student experience and outcomes. It provides a new lens for viewing student experience at university, highlighting the multifacetedness of this phenomenon and the diversity of possible patterns of student behavior. Based on the conceptual model, a ten-dimensional scale measuring student behavior was developed and validated through mixed-method research with an exploratory sequential design. The following dimensions of student behavior were identified: 1) interaction with course content in class; 2) persistence; 3) self-learning; 4) irresponsible learning behavior; 5) active learning; 6) friendship; 7) study collaboration; 8) obedience; 9) creating a positive self-image; and 10) extracurricular involvement. To develop a survey instrument, we utilized semistructured in-depth interviews with Russian students (n = 119). In the quantitative phase of the study, based on the survey (n = 1,253) carried out at seven highly selective Russian universities, we tested the reliability and validity of the ten-dimensional scale. To test construct-related validity, we utilized the self-determination theory developed by Ryan and Deci and a short version of the Academic Motivation Scale developed by Gordeeva, Sychev, and Osin for the Russian educational context. Our findings are in line with assumptions of self-determination theory and the results of previous studies and can be considered evidence of construct validity. The directions for further development of the methodological approach and its practical implications are discussed.

## 1. Introduction

Many researchers and practitioners in higher education have emphasized the crucial role of student experience at universities in developing students' knowledge and skills and their professional and personal growth [1–6]. Universities worldwide have begun to consider the enhancement of student experience as one of their key strategic priorities [7,8]. However, despite the wide academic discussion on this concept and highlighting its importance for

**Data Availability Statement:** All relevant data are within the manuscript and its Supporting Information files.

**Funding:** Support from the Basic Research Program of the National Research University

Higher School of Economics (HSE University) is gratefully acknowledged. The funder had no role in the study design, data collection and analysis, decision to publish, or preparation of the manuscript.

**Competing interests:** The authors have declared that no competing interests exist.

universities' strategic planning, student experience still has not received proper theoretical elaboration [9–12]. Attempts to define it are inconsistent and frequently limited to listing various aspects of the university experience [4,8,13–15].

Many studies have focused on certain aspects of student experience: student engagement [1,16–18], extracurricular involvement [19–21], time use [22–24], student–faculty interactions [25], student networking [26], academic misconduct and disengagement [27–29], student satisfaction [30–32], and school-to-university transition [33,34]. However, these investigations have yielded fragmentary knowledge about the student experience due to limitations in (1) comparing the results of various studies on student experience aspects and (2) utilizing the findings from this large amount of research to improve not just one or several aspects of the educational process and university environment at certain higher education institutions but also to develop an integrated educational policy that takes into account the complex combinations and interrelations among different aspects of student experience. Therefore, the use of "student experience" as an umbrella term that covers multiple separate concepts related to different aspects of university life restricts our capacity to improve student development. It is necessary to develop a complex multidimensional concept and methodology for its measurement, which can capture not only different aspects of student experience but also the interrelations between them.

The word "experience" takes the form of a noun and verb and includes the following meanings: 1) actions of the person that lead to certain results and changes in their personality in the form of new knowledge, skills, feelings, ideas about the world, attitudes, etc.; and 2) something that has occurred and influenced a person [35,36]. Using the phrase student experience, we limit a person's actions and what happens to them to the period of their higher education. The existing higher education studies mostly emphasize the second part of this definition, i.e., the responsibility and important role of higher education institutions in shaping and enriching this experience, including teaching practices, facilities, student services, and other university conditions and opportunities, which are considered important for student growth and can be utilized for the assessment of higher education quality [5,17,37,38]. However, they mostly ignore the influence of individual characteristics that can affect relationships between the environment, experiences, and outcomes at university. Previous research has shown that students' characteristics correlate with the main aspects of the student experience [39–45]. Hence, previous conceptions related to student experience are limited in explaining the mechanisms of how various aspects of the university environment and their combinations lead to certain outcomes for student development and why the experience of students from one educational program (i.e., in the same environment) can vary significantly.

We argue that not only the environmental aspects of student experience but also the *behavior* of students in university settings, which is affected by the combinations and interrelations of individual and environmental characteristics, is essential for student growth. However, little attention has been given to the plurality of student behavioral patterns at universities. As a rule, to examine what elements of the university environment are positive, studies have focused on a single aspect of student behavior (such as participation in extracurricular activities or student misconduct) without considering it in combination with other aspects of student behavior. Such methodological approaches limit the capacity of educators to improve educational outcomes by reinforcing desired student behaviors by changing elements of the learning environment.

The present study proposes the multidimensional concept of student behavior as a crucial element of student experience and develops and validates the survey instrument for its measurement.

## 2. Conceptualization of student behavior at university

### 2.1. Relevant concepts and theoretical frameworks

To conceptualize student behavior at university, we employ three types of theoretical and empirical literature: 1) theoretical papers explaining how learning occurs, 2) papers exploring various aspects of student experience and behavior, and 3) publications attempting to construct student typologies. A brief overview of these types of publications is given below.

**Paradigms explaining how learning occurs.** Two competing paradigms, related to student behavior but not their cognitive processing, were developed in educational research to explain how learning occurs and how academic outcomes can be improved: behaviorism and constructivism [46]. Behaviorism focuses on the *behavior* of students as a result of environmental stimuli and consequences and proposes that learning occurs in the process of operant conditioning [47,48]. Constructivism emphasizes the importance of communication between students and instructors and argues that learning occurs in the process of social interaction [49]. According to behaviorism, the role of an instructor is to prepare conditions and organize stimuli (course materials, assignments, assessments, etc.) in such a way as to reinforce desired behavior and accelerate the learning process [47,48]. In contrast, constructivism assigns an active role to students in shaping their learning experiences, setting educational goals, and achieving them through meaningful interactions at university [49–51].

Despite the concept of student behavior being close to the behavioristic paradigm in education, we suggest that ideas of constructivism are also productive for our conceptual model and can be utilized for conceiving some dimensions of student behavior, especially aspects related to communication with instructors and other students.

**Aspects of student experience and behavior.** One of the most discussed concepts is student engagement. Despite student engagement being an umbrella term that covers multiple definitions and methodological approaches [17,52], in the current research, we focus on the framework suggested by Kuh [18,53] and McCormick [17,54] and based on the ideas of Astin [1] and Pace [16]. This approach is of interest to our conception of student behavior for two reasons. First, it is considered to be an environmental one because the effect of the learning environment on the behavior and development of the person is highlighted [55]. Second, it focuses on observable acts of behavior at university and ignores cognitive processes and student efforts, which are hidden from view [52,56]. These two features of this student engagement approach are closely related to the behavioristic paradigm of learning at university, which is important for our conceptualization of student behavior. However, the student engagement approach prioritizes active learning practices that are more in the tradition of constructivism. Therefore, some researchers consider this approach constructivist [5].

The founders and supporters of the concept of student engagement argue that students acquire knowledge and skills through their activities at university and that a university can make this process more intensive and productive by providing opportunities for participation in various in-class and out-of-class activities as well as practices, which are positively correlated with academic outcomes and student growth [53,57,58]. The list of these practices is based on the research of Chickering and Gamson [59], who identified "seven principles of good practices" through the analysis of empirical research on the determinants of student success. These seven principles became a starting point for the development of survey instruments measuring student engagement [17,60]. The student engagement approach emphasizes those aspects of student behavior that are related to participation in active learning practices and can be considered evidence of the good quality of education at university.

The other aspect of student behavior at university is related to the social connection of students to classmates and faculty. The importance of this component of student life was

emphasized in the work of Tinto [61,62], who developed a framework explaining student dropout from university. He proposed that student dropout is caused by a low level of academic and social integration at university. Academic integration is related to students' academic performance, intellectual development, and experience in academic settings, whereas social integration refers to social connections and the presence of positive relationships with peers [61,63,64]. Tinto suggested that a high level of social integration can sometimes compensate for a low level of academic integration at university [65].

In addition, some researchers have identified several features of student behavior related to conformity and social desirability. Macfarlane [66] argued that the discourse of performativity resulted in phenomena such as student *presenteeism* and *learnerism*. Student presenteeism is defined as a tendency to invest an enormous amount of time in learning and demonstrate hard work and diligence despite feeling unwell or exhausted, even when these efforts are not effective. Learnerism involves incorporating characteristics of student behavior into the assessment criteria that force students to behave in certain ways, such as demonstrating interest and activity, asking questions to instructors, and participating in class discussions. The focus shifts from meaningful contributions to class activities to demonstrating socially desired behavior to be approved by instructors.

Finally, many researchers explore student involvement in extracurricular activities as a crucial element of student experience, which positively affects student growth and future career prospects [20,21,67,68]. Although extracurricular involvement is defined as student participation in university activities separate from the primary curriculum and not related to obtaining a degree [69], it might be an effective means for student retention and adaptation to university life [21,70]. Some researchers propose, however, that excessive extracurricular activities may have negative consequences, such as stress and anxiety, due to a high workload [71]. Therefore, we suggest that extracurricular activities constitute an important dimension of student behavior that should be considered in combination with other aspects.

**Student typologies.** There have been several attempts to construct comprehensive typologies of students based on several aspects of their experience and behavior at university. There are several typologies of students created on the basis of student time allocation between curricular and extracurricular activities at university and their life outside university [23,24,72]. Some researchers have introduced new concepts to systematize student behavior patterns [73,74]. For example, Clark and Trow [73] suggested the concept of student subcultures and dividing all students into groups according to their ideas, interests, attitudes towards university, and relationships with other students. In a recent book, Fischman and Gardner [74] proposed the concept of mental models to conceptualize student experience. This concept is used to capture students' ideas about what their learning process at the university should look like, how to use additional opportunities, and how to realize themselves within university settings. Although these conceptualizations illuminate some critical aspects of student life at university and attempt to systemize and capture differences in the student body, they have serious limitations. First, these typologies are highly sensitive to national and institutional contexts. Second, the attempts to reduce the plurality of student behavior patterns to a limited number of types lead to the oversimplification of student behavior, resulting in severe restrictions of these conceptual models for measuring and explaining student experience and behavior.

In summary, none of the existing conceptualizations can capture and explain the diversity of student behavior at university. The concepts and theoretical approaches developed in previous research either focus on the narrow aspects of student behavior and experience at university or oversimplify these phenomena, limiting the potential of using these approaches for research and practical implementation. The current study aims to overcome these limitations by developing a conceptual approach for obtaining a deeper understanding and measurement of diverse student behaviors at university.

## 2.2. Definition and five dimensions of student behavior at university

**Definition of student behavior.** Following the behaviorist paradigm, we define student behavior at university as a set of stable behavioral acts occurring in response to stimuli in the university environment that vary depending on the personal characteristics of the students. This definition has three important features that should be discussed in more detail. First, our conception emphasizes the role of the university and the characteristics of the program environment (faculty characteristics and course requirements, classmates, and facilities) in shaping student behavior. The effect of the educational environment on student behavior is proposed by the behaviorist paradigm [47,48] and has been demonstrated in many empirical studies [2,3,6].

Second, this definition highlights the diversity of students' behavioral responses to similar stimuli from the same educational program. The variety of student behavioral responses can be explained by differences in students' personal characteristics (previous background, motivation, psychological characteristics, etc.). This phenomenon has also been demonstrated in empirical research [39,75,76].

Third, we focus more on repeated behavioral responses because we consider them prerequisites of student growth and academic outcomes, whereas single behavioral responses may have nonsignificant or unstable effects on the development of students. Similar to the medical literature demonstrating the importance of healthy habits and adherence to therapies for success [77,78], we argue that regular behavior and study habits at university are crucial determinants of student growth.

**Five dimensions of student behavior.** Based on the review of the relevant concepts and approaches presented above, we suggest five dimensions of student behavior: 1) academic diligence; 2) active learning; 3) social integration; 4) conformity behavior; and 5) extracurricular involvement. Fig 1 presents the scheme of the conceptual model, where the concept of student behavior is located at the intersection of the characteristics of the university environment and the personal characteristics of students.

Academic diligence can be defined as student activities and efforts involving independent interaction with the learning material to achieve educational goals that are not associated with social communication with other students or instructors. These activities include paying attention in class, investing time and effort in learning at home, doing activities to deepen knowledge, forcing themselves to study, and organizing effective course work. We separate these activities and efforts from the cognitive and mental processes, such as thoughts, perceptions, emotions, and memorization strategies, as well as the peculiarities of knowledge and skill acquisition, decision-making, observing the facts and behavior of others, analyzing causal links, and making conclusions. These cognitive and mental processes are important in explaining student development and academic outcomes, but they cannot be considered behavioral acts. Therefore, to conceptualize this dimension of student behavior, we employ the theoretical assumptions of behaviorism.

The second dimension of student behavior at university is active learning, which can be defined based on previous research on student engagement [53,57]. We suggest that active learning involves student participation in educationally purposeful practices in class, which involves social interactions with instructors and other students. Active learning includes such behavioral patterns as active participation in class activities, group work, discussions, and dialogs with instructors and others. As this definition suggests the importance of social interactions, it is closer to assumptions of constructivism.

Both academic diligence and active learning are related to the curricular work of students. However, one of the main university functions is socialization. Therefore, it is important to

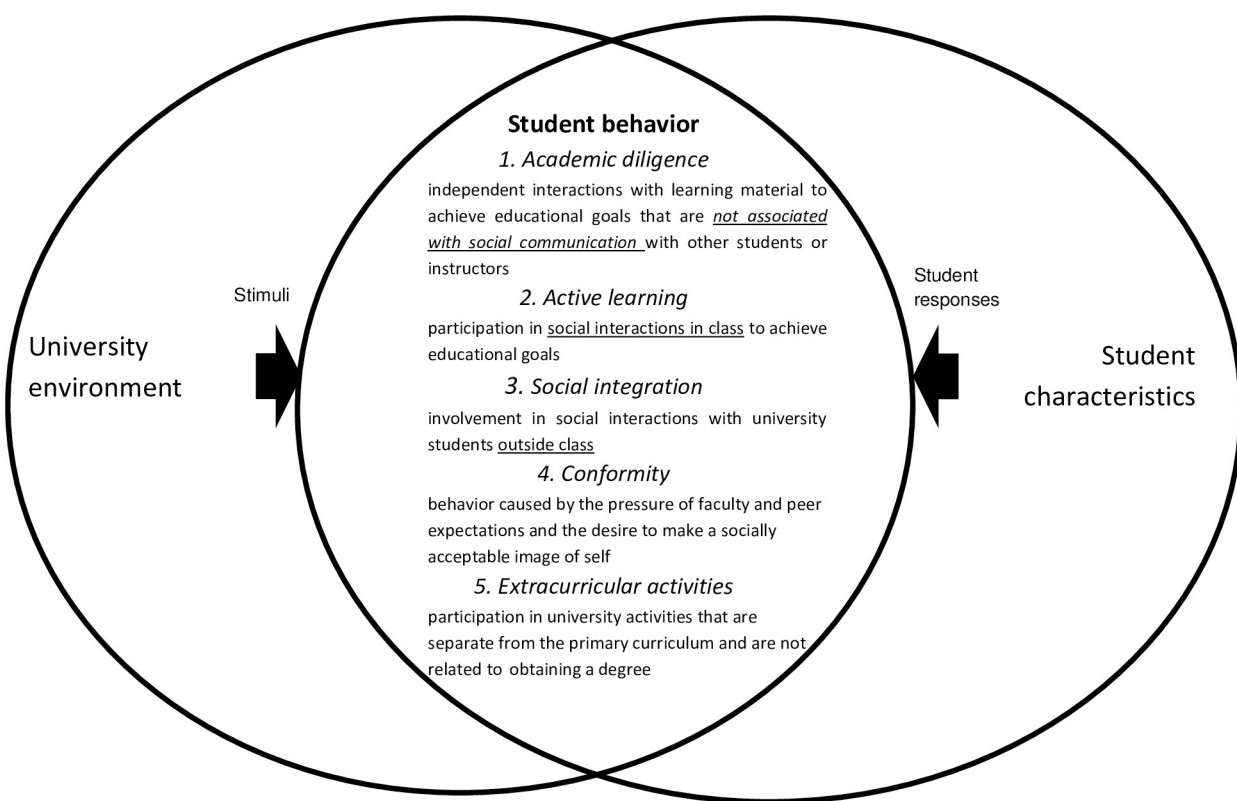

**Fig 1. The conceptual model of student behavior at university.**

identify behavior dimensions related to social aspects. In our conceptualization, we proposed two dimensions related to social bounds and group effects, which can occur in university settings: social integration and conformity behavior. Social integration includes the behavior of students related to building friendly relationships with other students at university and the use of these social bonds to reach educational goals and participate in collaborative study. The social integration dimension is close to the concept of social capital [79,80], which can accumulate during university studies and be used in university settings to achieve educational and personal goals. Previous research has demonstrated that social integration is correlated with student progress and growth at university [60,81,82].

However, the reverse side of the social connectedness and social nature of education at university is conformity. We define conformity behavior as a change in behavior caused by the pressure of faculty and peer expectations and the desire to make a socially acceptable image of self. We hypothesize that conformity behavior is a negative phenomenon that limits the effectiveness of learning at university, which has been partly demonstrated in previous studies [65,83]. We also expect that one type of conformity behavior is associated with students' desire to meet *faculty* expectations about 'a good student', whereas another type is associated with the desire to be 'cool' among *peers*. The second type of conformity behavior is based on the publication dedicated to creating a positive image of themselves and peer effects in educational settings [84–88]. For example, Jackson and Dempster [84] presented gender enactment by demonstrating differences in the learning behavior of men and women students. They reported that students described effortless achievement as 'cool' and 'popular' and associated it with masculinity, whereas hard work was described as 'uncool' and associated with femininity.

As students who visibly prioritize hard work are frequently labeled 'nerd' and 'swots' and are often bullied, some 'cool' students may try to avoid hard work to maintain their status [84,88].

Finally, we identify extracurricular activities as one of the behavioral dimensions that is in line with previous conceptual and empirical studies. This aspect is important because of its effect on academic outcomes, student growth, and well-being. The exploration of different combinations of these activities with curricular work is also productive in capturing the diverse patterns of student behavior at university.

The conceptual model is grounded in the behaviorist and constructivist paradigms explaining how learning occurs and is informed by the results of previous research on different aspects of student experience at university. The combination of different theoretical approaches with empirical evidence allows us to provide a productive methodological approach for exploring patterns of diverse student behaviors at university and their combinations. To develop a survey instrument measuring student behavior at university, we conducted mixed-method empirical research with an exploratory sequential design.

## 3. Materials and methods

### 3.1. Development of the instrument

To develop the instrument and generate items for each of the five aspects of student behavior at university, semistructured interviews were conducted. First-year students (n = 119) enrolled in eight highly selective Russian universities in 2022 took part in this phase of the research. Highly selective universities in Russia represent the group of higher education institutions that have the highest chances of attracting the most talented students. To be admitted to university, Russian applicants must pass the Unified State Exam (USE). Universities select applicants based on their USE scores. The applicants with the highest scores can be admitted to the most prestigious universities, and the quality of admission at Russian universities is estimated by the average USE scores of admitted students. We consider universities in the highest quartile by the average USE score of admitted students to be highly selective. Therefore, our participants, who are represented by students from highly selective universities, may be more talented and motivated to study at university than the general population of Russian students.

S1 File presents the procedures for the collection and analysis of the interview data and the generation of initial items of the student behavior questionnaire. S2 File presents the final version of the developed survey instrument in both English and Russian. Owing to the national context of the study, the initial items were formulated in Russian. The back-translation procedure was used by both authors to ensure the equivalency of the Russian and English versions. The pool of initial items included 20 items measuring academic diligence, 11 items measuring active learning, 11 items measuring social integration, 7 items measuring conformity behavior, and 6 items measuring extracurricular involvement. For each item, the students were asked to indicate how often they performed the activities described in the item by choosing one of the following options: "never", "rarely", "sometimes", "somewhat often", or "very often". This scale reflects our idea that, by the concept of student behavior, we mean *repeated* behavioral acts. Thus, we believe that the more often a student behaves in certain ways in university settings, the more prominent a certain behavioral pattern is.

### 3.2. Quantitative data

Data for this research were collected during a longitudinal study titled "Educational behavior and success of university students in Russia" launched in the fall of 2022 in seven Russian universities (one of the eight universities participating in the qualitative phase did not participate in the survey). This study aims to explore changes in the behavior and academic performance of students

enrolled in four-year undergraduate programs in 2022 during the whole period of their study at universities. The Ethics Committee of HSE University, Russia, approved this study on September 19, 2022. Between October 11, 2022, and December 26, 2022, 3,353 first-year students participated in the first wave of the longitudinal study, with a response rate (RR) of 66%. Each respondent was provided with an informed consent form, which was presented on the first screen of the online survey. To proceed with the online questionnaire, the respondents were required to click the button "To sign the form". All respondents were of age, allowing them to sign the consent form by themselves according to Russian legislation (15 years or older). Then, they were asked to take a standardized test on critical thinking and to respond to questions from an online survey about their demographics, family characteristics, educational background, and school-university transition. The developed instrument measuring student behavior at university was included in the longitudinal survey as a part of the questionnaire for the second wave of the survey, which was conducted from October 26, 2023, to December 13, 2023. This questionnaire also included questions about well-being, motivation, self-reported assessment of skills improvement, time use, and satisfaction with the educational programs and the university. A total of 1,679 students from those who participated in the first wave completed the second wave of the survey (RR = 33%).

The convenience sampling method, which included two stages, was utilized. In the first stage, we randomly selected undergraduate programs (n = 131) at each university such that at least one program from each field of study (mathematics and science, engineering, medicine, agriculture, social sciences, humanities, and arts) was included in the research. In the second stage, each student from the selected programs was invited to participate in the research by email. Convenience sampling is a nonrandom sampling technique involving respondents who are "convenient" to a researcher (for example, recruiting people in the street, in a workplace, or on the internet). In our research, we recruited students who used email services, opened emails with invitations and were willing to participate in the survey. The recruiting students through emails and the volunteer nature of their participation can lead to bias in the data discussed in the Limitations section. The data collection procedures were the same for all the participating universities.

The current study utilizes data collected through these two waves of online surveys of students: demographics and family characteristics from the first wave and questions measuring student behavior and motivation from the second wave. For the current research, we selected students (n = 1620) enrolled only in programs in four fields of study (mathematics and science, engineering, social sciences, and humanities) for two reasons. First, these fields are more common in Russian universities, and these programs have fewer peculiarities related to specific professions (in contrast with medicine, agriculture, and arts). Second, we received only a few responses from students enrolled in other fields.

The following variables contained missing values: sex (0.9%, 14 missing values), family income (15.9%, 257 missing values), and items measuring student behavior (up to 6.5%, 106 missing values). In total, the 367 observations with missing data on the variables used in the analysis were removed. The missing data were not replaced. The final sample included 1,253 respondents. Tables 1–3 in S1 Table present the descriptive statistics of the final sample. Fifty-three percent of the sample are male, 59% are from engineering majors, 16% are from mathematics and science, 16% are from the humanities, and 9% are from the social sciences. Most of the sample (70%) consists of state-subsidized students. Sixty-four percent of the sample are from middle- and high-income families.

### 3.3. Data handling and statistical analyses

Data analyses were performed via SPSS 20.0 and RStudio 2019 (version 1.2.5001) [89]. Our analytic strategy had four steps. In the first step, we ran exploratory factor analysis (EFA) for

each of the five sets of items developed based on the theoretical approach and interview data. Principal component analysis (PCA) was applied to identify the factor structure and to determine the number of factors measuring each aspect of student behavior. In the second step, the internal reliability of the scales was estimated by calculating Cronbach's α. In the third step, confirmatory factor analysis (CFA) was carried out to assess the evidence based on the internal structure of the questionnaire [90]. As we employ ordinal scales in items measuring student behavior at university, the diagonally weighted least squares (DWLS) estimator was used. The model fit was evaluated based on the following fit indices: chi-square, the comparative fit index (CFI), the Tucker–Lewis index (TLI), the root mean squared error of approximation (RMSEA), and the standardized root mean square residual (SRMR). According to Schreiber et al. [91], Browne and Cudeck [92], and Hu and Bentler [93], CFI and TLI values higher than 0.90, RMSEA values less than 0.06, and SRMR values less than 0.08 can be considered evidence of an acceptable fit.

Finally, to assess the evidence based on relationships with other variables, we tested the construct-related validity of the instrument [90]. We studied the relationships between student behavior and academic motivation, which are similar constructs. We specified a structural model in which ten factors of student behavior are dependent variables and four factors measuring motivation are independent variables that predict student behavior at university. The following control variables were included in the proposed structural model: 1) sex (0 –male; 1 –female); 2) field of study; 3) form of financing (0 –fee-based study; 1– state-subsidized study); and 4) family income group (0 –low- and middle-income group; 1 –high-income group).

## 3.4. Evidence based on the relation to other constructs

To test construct-related validity, we developed a conceptual model of the relationship between motivation and student behavior, which was based on the assumptions of self-determination theory. Deci and Ryan suggested that human behavior is driven by intrinsic and extrinsic motives [94].

**Main assumptions of self-determination theory.** According to Deci and Ryan [95], intrinsic motivation is related to the desire to perform some activities for inherent satisfaction, fun, challenge, or extending capacities (innate needs for competence), whereas extrinsic motives drive people to act for some separable outcomes, such as avoiding punishment or receiving an external reward. To measure motivation, the short version of the Academic Motivation Scale, which was developed and validated in the Russian educational context by Gordeeva, Sychev, and Osin [96], was employed. This instrument suggests items to measure two types of intrinsic motivation (intrinsic cognition and achievement) and two types of extrinsic motivation (introjected and external regulation). Intrinsic motivation is divided into types on the basis of the needs that students satisfy through certain behaviors: cognition, achievement, or personal growth [96]. In the present research, we utilized only two types related to cognition and achievement. The intrinsic cognition scale measures the students' desire to learn new things and understand the subject, which is associated with experiencing interest and pleasure from the learning process itself [96]. Achievement motivation is defined as the desire to achieve high academic outcomes and to feel pleasure from the process of solving complicated problems [96].

The types of extrinsic motivation are differentiated by the degree of autonomy. Deci and Ryan [95] identified four types of extrinsic motivation: integrated, identified, introjected, and external. In the present research, we measure only two types, i.e., introjected and external, which are considered less autonomous regulations. External regulation is defined as motives that lead to the least autonomous behavior driven by the need to satisfy external demand or

reward contingency [97]. Introjected regulation is more autonomous than external regulation. This suggests that individuals act in such a way that they avoid guilt or anxiety and attain ego enhancements [98]. Although this behavior is internally driven, it has an external perceived locus of causality as well as external regulation. In educational contexts, introjected regulation was positively related to investing more effort by students, but it was also related to anxiety and coping more poorly with failure [97]; externally regulated students tended to put less effort and blame others for negative outcomes [97]. Benware and Deci [99] reported positive correlations between engaging in active learning, conceptual understanding, and intrinsic motivation. Kusurkar et al. [100] reported that intrinsic motivation correlates with persistence, willingness to sacrifice, and readiness to start in medical students; introjected regulation has only modestly significant correlations with willingness to sacrifice and readiness to start, whereas external regulation negatively correlates with persistence and has no significant links with willingness to sacrifice and readiness to start [100]. Maloshonok [45] reported that extracurricular activities such as research work are positively correlated with motives related to achievement (e.g., developing professional skills, obtaining new knowledge, and broadening intellectual horizons).

**Conceptual model of the relationship between motivation and student behavior for testing construct validity.** Based on the theoretical assumptions and the results of previous research on these types of motivation, we suggest the following relationships between the four types of motivation and the five dimensions of student behavior. We expect that *intrinsic cognition* is positively related to academic diligence and active learning, as well as enhancing student integration and involvement in extracurricular activities. *Achievement* motivation is positively associated with academic diligence, active learning, and extracurricular involvement. In terms of external motivation, we expect that *introjected regulation* is positively linked with academic diligence and conformity behavior, whereas *external regulation* is negatively associated with academic diligence, active learning, conformity behavior and extracurricular involvement.

## 4. Results

In this section, we demonstrate the psychometric properties of the instrument for measuring student behavior. First, we identify the factor structure of each aspect of student behavior that was assumed based on the literature review and quantitative phase of the research. This analysis allows us to identify a certain number of student behavior dimensions. Second, to demonstrate the reliability of the instrument, we investigate the internal structure of the dimensions and their interrelations with each other. Finally, to test the construct validity of the instrument, we explore the relationships between student behavior dimensions and other constructs. For this purpose, we utilize the self-determination theory developed by Ryan and Deci [95] and a short version of the Academic Motivation Scale developed by Gordeeva, Sychev, and Osin [96] for the Russian educational context. The observed correlations between student behavior dimensions and different types of motivation that are in line with the assumptions of self-determination theory are evidence of the construct validity of the instrument.

### 4.1. Exploratory factor analysis and internal validity

To identify the factor structure of each aspect of student behavior, we conducted EFA.

*1. Academic diligence.* The mean values of all the items ranged from 1.43 to 3.95 (standard deviation (SD): 0.80–1.23). The results of the Kaiser–Meyer–Olkin test demonstrated sample adequacy (KMO = 0.89). The results of Bartlett's test of sphericity ($\chi^2$ (190) = 9594.32, $p < 0.001$) verified that the correlations between items were appropriate for factor analysis.

Four factors with eigenvalues above 1.0 were extracted. Based on the analysis of items with high factor loadings, these factors were labeled as follows: 1) interaction with course content in class; 2) persistence; 3) self-learning; and 4) irresponsible learning behavior.

*2. Active learning.* The mean values of all the items ranged from 1.94 to 3.69 (SD: 1.00–1.22). The results of the KMO test demonstrated sample adequacy (0.90). The results of Bartlett's test ($\chi 2$ (55) = 5835.45, df = 55, p < 0.001) verified that the correlations between items were appropriate for factor analysis. Only one factor with an eigenvalue above 1.0 was extracted.

*3. Social integration.* The mean values of all the items ranged from 2.30 to 4.22 (standard deviation (SD): 0.97–1.26). The results of the KMO test demonstrated sample adequacy (KMO = 0.89). The results of Bartlett's test of sphericity ($\chi 2$ (55) = 6587.76, p < 0.001) verified that the correlations between items were appropriate for factor analysis. Two factors with eigenvalues above 1.0 were extracted. Based on the analysis of items with high factor loadings, these factors were labeled as follows: 1) friendship and 2) study collaboration.

*4. Conformity behavior.* The mean values of all the items ranged from 1.72 to 3.61 (standard deviation (SD): 1.00–1.19). The KMO value is 0.72, which is considered a middling value but is still acceptable [101]. The results of Bartlett's test of sphericity ($\chi 2$ (21) = 1630.72, p < 0.001) verified that the correlations between items were appropriate for factor analysis. Two factors with eigenvalues above 1.0 were extracted. Based on the analysis of items with high factor loadings, these factors were labeled as follows: 1) obedience and 2) creating a positive self-image.

*5. Extracurricular involvement.* The mean values of all the items ranged from 1.96 to 2.60 (SD: 1.22–1.32). The results of the KMO test demonstrated sample adequacy (0.90). The results of Bartlett's test ($\chi 2$ (15) = 5267.20, p < 0.001) verified that the correlations between items were appropriate for factor analysis. Only one factor with an eigenvalue above 1.0 was extracted.

Based on the preliminary EFA, we identified a 10-factor structure of the initial pool of items. We then conducted EFA and reliability analysis for each extracted factor. The factor loadings and explained total variance are presented in Table 1. Cronbach's alpha was used to test the internal consistency reliability of all the factors extracted via EFA. The coefficients for 8 of the 10 factors exceed the threshold of 0.7 [102] (see Table 1).

## 4.2. Evidence based on internal structure

Considering the results of the EFA, we propose a ten-dimensional model of student behavior at university that includes the following factors: 1) interaction with course content in class; 2) persistence; 3) self-learning; 4) irresponsible learning behavior; 5) active learning; 6) friendship at university; 7) study collaboration; 8) obedience; 9) creating a positive self-image; and 10) extracurricular involvement. The model assumes that the underlying ten factors of student behavior correlate with each other. One item measuring active learning (*"acted in a play"*) was deleted from the final model to improve model fit. The CFA of the final 54 items measuring 10 dimensions of student behavior at university demonstrated an acceptable fit to the data: $\chi 2$ (1332) = 6068.14, p < 0.001, CFI = 0.939; TLI = 0.935, RMSEA = 0.053, and SRMR = 0.062. All the items have significant factor loadings on their underlying constructs. The factor loadings are presented in S2 Table. The correlation matrix for the 54 items of the questionnaire is presented in S3 Table.

## 4.3. Evidence based on the relation to other constructs

The proposed structural model of correlation between motivation and student behavior dimensions demonstrated acceptable fit to the data: $\chi 2$ (2540) = 9326.14, p < 0.001,

**Table 1. Results of the EFA (PCA) and reliability analysis of the ten factors of student behavior at university.**

| Factors | Items | Factor loadings | Total variance explained | Cronbach's alpha |
|---|---|---|---|---|
| 1. Academic diligence | | | | |
| 1.1. Interaction with course content in class | 4 | 0.70–0.77 | 53.53% | 0.71 |
| 1.2. Persistence | 6 | 0.58–0.72 | 46.11% | 0.76 |
| 1.3. Self-learning | 4 | 0.78–0.82 | 64.13% | 0.81 |
| 1.4. Irresponsible learning behavior | 6 | 0.64–0.77 | 50.51% | 0.80 |
| 2. Active learning | 11 | 0.57–0.76 | 44.12% | 0.87 |
| 3. Social integration | | | | |
| 3.1. Friendship | 6 | 0.66–0.76 | 52.99% | 0.82 |
| 3.2. Study collaboration | 5 | 0.68–0.80 | 54.75% | 0.80 |
| 4. Conformity behavior | | | | |
| 4.1. Obedience | 3 | 0.70–0.80 | 54.09% | 0.57 |
| 4.2. Creating a positive image of self | 4 | 0.64–0.77 | 51.89% | 0.69 |
| 5. Extracurricular involvement | 6 | 0.70–0.88 | 66.59% | 0.90 |

CFI = 0.941; TLI = 0.936, RMSEA = 0.046, and SRMR = 0.055. Most of the structural paths supported the theoretically assumed relationships between latent variables (Tables 2–4). Intrinsic cognition is positively associated with interaction with course content in class (B = 0.20, $p < 0.05$), active learning (B = 0.17, $p < 0.01$), student friendship (B = 0.21, $p < 0.05$), and study collaboration (B = 0.30, $p < 0.001$). Achievement motivation is positively correlated with persistence (B = 0.21, $p < 0.05$), self-learning (B = 0.57, $p < 0.001$), active learning (B = 0.14, $p < 0.01$), and extracurricular involvement (B = 0.19, $p < 0.01$). Introjected regulation has a strong positive link with all dimensions of academic diligence: interaction with course content in class (B = 0.42, $p < 0.001$), persistence (B = 0.67, $p < 0.001$), self-learning (B = 0.29, $p < 0.01$), and irresponsible learning behavior (B = -0.37, $p < 0.01$). This type of external motivation is also strongly correlated with obedience (B = 0.65, $p < 0.001$). In contrast, external regulation is negatively associated with course content in class (B = -0.38, $p < 0.01$), persistence (B = -0.66, $p < 0.01$), self-learning (B = -0.36, $p < 0.05$), and obedience (B = -0.51, $p < 0.01$) and

**Table 2. Regression coefficients (B (SE)) for the model predicting academic diligence dimensions.**

| | Interaction with course content in class | Persistence | Self-learning | Irresponsible learning behavior |
|---|---|---|---|---|
| Female | 0.05 (0.04) | 0.21*** (0.04) | -0.15** (0.05) | -0.03 (0.03) |
| Field of study (ref.–mathematics and science) | | | | |
| Engineering | -0.06 (0.07) | 0.05 (0.08) | 0.08 (0.09) | -0.05 (0.05) |
| Social sciences | 0.01 (0.09) | -0.03 (0.11) | -0.11 (0.12) | 0.07 (0.07) |
| Humanities | 0.07 (0.08) | -0.03 (0.10) | 0.02 (0.11) | -0.03 (0.07) |
| Motivation | | | | |
| Intrinsic cognition | 0.20* (0.10) | -0.03 (0.14) | -0.19 (0.13) | 0.03 (0.10) |
| Achievement | 0.08 (0.07) | 0.21* (0.10) | 0.57*** (0.10) | -0.08 (0.07) |
| Introjected regulation | 0.42*** (0.12) | 0.67*** (0.18) | 0.29* (0.13) | -0.37** (0.12) |
| External regulation | -0.38** (0.13) | -0.66** (0.20) | -0.36* (0.15) | 0.54*** (0.14) |
| State-subsidized | 0.07 (0.04) | -0.01 (0.05) | -0.04 (0.05) | 0.10** (0.03) |
| High family income | 0.02 (0.03) | 0.04 (0.03) | -0.01 (0.04) | -0.01 (0.02) |

Note

*: $p < 0.05$

**: $p < 0.01$

***: $p < 0.001$.

**Table 3. Regression coefficients for the model predicting the active learning and social integration dimensions.**

| | Active learning | Student friendship | Study collaboration |
|---|---|---|---|
| Female | -0.13*** (0.03) | -0.05 (0.04) | 0.03 (0.04) |
| Field of study (ref.–mathematics and science) | | | |
| Engineering | -0.13* (0.05) | 0.07 (0.08) | 0.12 (0.08) |
| Social sciences | 0.35*** (0.08) | 0.30** (0.11) | 0.41*** (0.11) |
| Humanities | 0.25*** (0.07) | 0.17 (0.10) | 0.11 (0.10) |
| Motivation | | | |
| Intrinsic cognition | 0.17** (0.06) | 0.21* (0.08) | 0.30*** (0.08) |
| Achievement | 0.14** (0.04) | 0.09 (0.06) | -0.02 (0.06) |
| Introjected regulation | 0.06 (0.05) | -0.01 (0.08) | 0.08 (0.07) |
| External regulation | -0.08 (0.06) | 0.08 (0.09) | -0.02 (0.09) |
| State-subsidized | 0.06 (0.03) | 0.01 (0.05) | 0.02 (0.05) |
| High family income | 0.05* (0.02) | 0.07* (0.03) | 0.11*** (0.03) |

Note

*: $p < 0.05$

**: $p < 0.01$

***: $p < 0.001$.

positively associated with irresponsible learning behavior (B = 0.54, $p < 0.001$). These findings are in line with the assumptions of self-determination theory and the results of previous studies [94,97,100] and can be considered evidence of construct validity.

We also observed that all ten factors of student behavior correlate with each other. The covariance matrix is presented in S4 Table.

## 5. Discussion

This study conceptualized student behavior at university and successfully developed and validated an instrument measuring this complex construct through a ten-dimensional scale. The

**Table 4. Regression coefficients for the model predicting conformity and extracurricular activities.**

| | Obedience | Creating a positive image of self | Extracurricular involvement |
|---|---|---|---|
| Female | 0.01 (0.05) | -0.04 (0.05) | -0.11* (0.04) |
| Field of study (ref.–mathematics and science) | | | |
| Engineering | 0.06 (0.09) | 0.14 (0.09) | -0.13 (0.08) |
| Social sciences | 0.20 (0.13) | 0.20 (0.12) | 0.05 (0.11) |
| Humanities | 0.23* (0.12) | 0.05 (0.11) | 0.11 (0.11) |
| Motivation | | | |
| Intrinsic cognition | -0.02 (0.14) | -0.11 (0.09) | 0.01 (0.08) |
| Achievement | 0.02 (0.10) | 0.02 (0.07) | 0.19** (0.06) |
| Introjected regulation | 0.65*** (0.16) | 0.06 (0.09) | 0.04 (0.08) |
| External regulation | -0.51** (0.18) | 0.12 (0.26) | -0.05 (0.09) |
| State-subsidized | 0.04 (0.05) | 0.02 (0.05) | -0.07 (0.05) |
| High family income | 0.05 (0.04) | -0.03 (0.04) | -0.04 (0.03) |

Note

*: $p < 0.05$

**: $p < 0.01$

***: $p < 0.001$

conceptual and measurement model is grounded in the behaviorist and constructivist paradigms explaining how learning occurs, as informed by the results of previous research on different aspects of the student experience at university. The combination of different theoretical approaches with empirical evidence allows us to provide a productive methodological approach for exploring patterns of diverse student behaviors at university and their combinations. It also overcomes limitations of previous theoretical approaches, which (1) suffer from insufficient theoretical elaboration, (2) propose measuring only single aspects of the student experience [19,21,25–27,29 and others], or (3) oversimplify the concept of the student experience at university by constructing a comprehensive typology of students through identifying only a small number (4–6) of groups [73,74].

In the conceptualization phase of the study, we proposed five aspects of student behavior at university, which were based on the analysis of previous research on student experience in higher education: 1) academic diligence; 2) active learning; 3) social integration; 4) conformity behavior; and 5) extracurricular activity. We assume that these aspects correlate with each other but nevertheless can be combined by students into many behavioral patterns.

## 5.1. Academic diligence

Academic diligence was defined as student activities and efforts aimed at achieving educational goals at university that cannot be directly observed. The identification of this aspect of student behavior allows us to respond to the criticism of the student engagement approach of ignoring nonobservable student effort and activities [52,56]. The qualitative phase of the research provides information for generating 20 items measuring different aspects of diligence. Through the analysis of the factor structure, we identified four dimensions of academic diligence: 1) interaction with course content in class; 2) persistence; 3) self-learning; and 4) irresponsible learning behavior. Interaction with course content measures student behavioral acts that take place during classes but are related to directly nonobservable activities and can be differentiated from the phenomenon of active learning. Active learning suggests social interactions in class, in which an instructor and students participate in dialogs with each other, whereas interaction with course content suggests that the instructor delivers course material and that students "receive" it by utilizing learning strategies such as note-taking, attentive listening, and relating new course material to what the student already knows. These activities should be marked as behavioral because we do not consider the results of the cognitive process and their specific features but only the fact that students invest effort in doing so.

Persistence is defined as a type of behavior related to overcoming difficulties and organizing conditions in such a way as to achieve educational goals. Persistence can be measured through indicators of hard work despite a loss of interest, planning one's own learning activities, and making efforts to make learning effective. This dimension of student behavior is closely related to the concept of self-regulated learning developed in the work of Zimmerman [103] and Bandura [104].

Self-learning is a behavior related to additional activities aimed at better understanding course materials. It is strongly associated with achievement motivation, as students believe that their extra effort can lead to greater academic outcomes. Achievement motivation is related to enjoyment from solving difficult tasks, which require more work than is required by courses. Hence, we can conclude that this correlation is in line with the theoretical assumptions of Deci and Ryan [95].

Irresponsible learning behavior is a dimension that should be considered the opposite of academic diligence. It captures two aspects of student experience covered in previous research: disengagement, proposed by Brint and Cantwell and defined as a lack of commitment to

curricular work in students' daily activities [29,P. 810], and academic misconduct [27,28]. Both constructs relate to the violation of the educational process requirements but in different ways. Disengagement is inactivity when needed, and academic misconduct involves activities that are considered "wrong" in academic settings. The analysis of the factor structure and validity of items, which measures both phenomena but is integrated into one scale, allows us to conclude that both phenomena can be considered one latent variable when the research aim is related to the measurement of academic diligence.

According to the results of structural equation analysis, academic diligence is associated with student motivation. We found strong positive associations of academic diligence dimensions with introjected regulation and strong negative correlations with external regulation, which are in line with previous research [97,100] and theoretical assumptions for the conceptualization of these two types of extrinsic motivation [98]. Interaction with course materials can also be driven by internal cognition motivation, while persistence and self-learning can be driven by achievement motivation. The moderate correlations between intrinsic motivation and academic diligence can be explained by the features of the Russian educational context, in which curricular work in class is characterized by a high prevalence of passive learning practices and a low share of active learning practices that encourage intrinsic motivation [45,99].

Taking into account the significance and directions of structural paths between latent variables, we conclude that the division of academic diligence into four separate dimensions of student behavior is justified and productive for capturing the variety of student behavior patterns at university, as it can be explained by both the personal characteristics of students and factors in the university environment. Overall, the analysis results support the reliability and validity of the items for measuring academic diligence.

### 5.2. Active learning

Active learning is defined as student participation in educationally purposeful practices in class, which involves the observed active behavior of students and their involvement in student–faculty interactions. These two features differentiate this dimension from academic diligence. Eleven items for measuring this aspect were generated based on previous research on student engagement [17,60] and the result of the qualitative phase of the research. The good internal consistency of these items was demonstrated based on the Cronbach's alpha coefficient and CFA. EFA demonstrated that the variance in the items can be explained by one factor. However, one item was removed from the analysis to improve the fit of the CFA model.

According to the structural model, active learning is driven by intrinsic motivation (intrinsic cognition and achievement), which is in line with the theoretical assumptions and findings of Benware and Deci [99], who demonstrated that students involved in active learning demonstrate more intrinsic motivation.

### 5.3. Social integration

According to our conceptual model, social integration includes the behavior of students related to building friendships with other students at university and the use of these social bonds to reach educational goals and participate in collaborative studies. The qualitative phase of the research provides information for generating eleven items measuring social integration. Through the analysis of the factor structure, we identified two dimensions of this type of behavior: friendship and study collaboration. Friendship suggests student involvement in close relationships with classmates, which are not directly related to study. Study collaboration implies participation in dialogs and group work with classmates on topics related to course materials. The two dimensions are differentiated by the nature of the questions around which

communication with classmates is built: those related to studies and those not related to studies. The items measuring both dimensions of social integration demonstrated good internal consistency. Both factors are correlated with intrinsic cognition. This finding is in line with those of previous studies dedicated to second language learners, which demonstrated a positive correlation between intrinsic motivation and willingness to communicate [105,106]. Another study demonstrated that social integration into the peer context is a crucial prerequisite for developing intrinsic motivation [107]. As our analysis reveals the positive association between social integration and intrinsic motivation, we can consider this as evidence of construct-related validity.

### 5.4. Conformity behavior

We define conformity behavior as "the dark side" of social integration within peer and faculty contexts and of the social nature of education at university. In our conceptual model, it is defined as a change in behavior caused by the pressure of faculty and peer expectations and the willingness to create a socially desired self-image. The identification of this behavioral dimension is supported by previous research demonstrating conformity in university students [65,83], as well as the results of our qualitative phase, in which students admitted that they try to adjust their behavior in class to meet faculty expectations, creating an image of "a good student". This dimension may have greater importance in cultures that assume high power distance between teachers and students and promote student compliance and teacher authority, as well as in classes with international students or ethnic minorities from those cultures.

In the semistructured interviews, many students described their willingness to act according to faculty expectations and a fear of contradicting them. However, our participants did not mention their willingness to affect their classmates' perceptions of them through behavioral acts. Nevertheless, we decided to develop items measuring this aspect of conformity and include them in the survey instrument. Through the analysis of the factor structure, we identified two dimensions of conformity behavior: obedience and creating a positive self-image. Obedience refers to behavior influenced by faculty expectations even if it negatively affects the learning process. Creating a positive self-image is related to behavior aimed at making a good impression on faculty and classmates. Although the CFA model demonstrates the internal consistency of these items, the Cronbach's alpha coefficient for both dimensions is close to 0.6–0.7, which is considered acceptable [108] but is lower than the cutoff point (0.7). Although this reliability can be considered sufficient for a preliminary analysis of student conformity, including group comparisons and testing correlations with other variables, further methodological work should be done to develop measurements with better reliability.

We find that obedience is positively correlated with introjected regulation and negatively correlated with external regulation, which is in line with our theoretical assumptions. However, the second conformity dimension was not found to be correlated with academic motivation. Considering the results of our analysis, we can conclude that the conformity behavior dimension is important for a deeper understanding of the variety of behavioral patterns at university and the factors influencing them. However, the types of conformity behavior and the items measuring this construct require further methodological elaboration to improve the psychometric properties of the instrument.

### 5.5. Extracurricular involvement

We define extracurricular involvement, in line with previous studies [68], as student participation in university activities that are separate from the primary curriculum and are not related to obtaining a degree. To develop items measuring this dimension, we utilized the results of

the qualitative phase of the study and previous research. We identified the most widespread extracurricular activities in Russian universities and chose those that match extracurricular activities in other countries. Six items demonstrating good reliability were developed. EFA demonstrated that the variance in the items can be explained by one factor. Extracurricular involvement positively correlates with achievement motivation, which is in line with our theoretical assumptions [96] and previous research [45] and can be considered evidence of construct-related validity.

## 6. Limitations

The current study has several limitations. First, the data are restricted by their convenience sampling method and the number and types of represented universities. The study was conducted in only highly selective universities, which attracted more talented and motivated students and had higher curricular requirements. Since we were restricted to first-year students from highly selective Russian universities in the qualitative phase of the study, some specific behavioral aspects of students from nonselective universities may have been missed and not reflected in the questionnaire. Convenience sampling methods in the quantitative phase may result in self-selection bias. We assume that more academically motivated students may take part in both waves of the longitudinal survey. Second, owing to the longitudinal design of the survey, we have many missing values, resulting in the removal of some cases from the final database and increasing possible response bias.

Third, the Russian system of higher education, where the current study was carried out, has several specific features, such as 1) a rigid curriculum and narrow specializations, 2) a high proportion of state-subsidized students who do not have to pay tuition fees for their studies, 3) the prevalence of passive learning practices in class, 4) a high curricular workload, and 5) low student involvement in extracurricular activities [45,109]. These features can affect the results of our research: the behavioral aspects identified, the items developed for their measurement, and the observed associations between academic motivation and student behavior. Therefore, the construct validity of this survey instrument must be evaluated and established not only in samples derived from other types of higher education institutions but also in a wider variety of national contexts.

Finally, the study tests construct-related validity only through the analysis of associations between student behavior dimensions and motivation. Further research should test the effects of different dimensions of student behavior on academic performance and different measures of student growth at university.

## 7. Conclusions and practical implications

We synthesized the results of various empirical studies on different aspects of student experience to develop an integrated theoretical framework that considers the multidimensional nature of student behavior at university and to develop an instrument that measures this variety of dimensions without oversimplifying the diversity of behavioral patterns at university. We also established support for the validity of a ten-dimensional instrument measuring student behavior at university, both as a whole and for its individual subscales. We argue that the concept of 'student behavior' is a promising alternative to the concept of 'student experience' in higher education research because it encompasses different aspects of student life and their interrelation with each other, explains the effects of environmental and personal characteristics on student performance and growth, and has practical implications through its measurement and management.

The main peculiarities of the student behavior approach are as follows. First, we focus not only on institutional and teaching practices that contribute to student development but also

consider student characteristics and the features that arise as a result of the interaction of their characteristics with certain stimuli from the educational environment. Second, we pay attention to those aspects of behavior that are not directly related to teaching practices and activities in the classroom. Unlike student experience and student engagement approaches, we pay much more attention to the behavioral acts that cannot be directly observed in the classroom since they do not imply obvious active actions; this is particularly relevant in universities with few or no "good" pedagogical practices that imply activity, such as those where many classes are held in a lecture format or where cultural traditions hinder student activity.

Although the instrument requires additional methodological work for its further validation in various national and institutional contexts, as well as item revision for the conformity behavior subscale, this instrument can be utilized in research and practical goals. The results of this research may be used in further academic and institutional research and can inform educational policy.

The ten-dimensional student behavior scale has the same advantages as student engagement and student experience approaches: 1) it focuses on the process indicators that allow universities to identify problems in the educational process in a timely manner (before they are observed as a decrease in academic outcomes), and 2) it is easier to use and has lower costs than approaches measuring student or alumni outcomes [37]. Moreover, our approach focuses not only on institutional and teaching practices but also on student characteristics and their effects on the effectiveness of these institutional and teaching practices. This allows us to expand possible practical means to enhance educational quality by identifying students requiring additional attention and the types of support that can help such students achieve better academic outcomes. Therefore, our approach does not provide a "panacea" list of good practices that signify good-quality education, as seen in student experience and student engagement approaches; this suggests more flexible ways to improve student outcomes that consider the characteristics of students and institutional and national contexts.

The student behavior scale can be utilized for the following practical purposes: 1) identification of negative behavioral patterns at university and providing data to develop a policy that overcomes these negative patterns to achieve higher academic outcomes; 2) assessment of the dynamics in the educational quality of the programs (even in universities and programs where active learning is not widespread or limited due to cultural peculiarities); 3) identification of at-risk students; 4) exploration of the effects of innovations related to the educational process on students; and 5) institutional exchange of practices that enhance positive behavior patterns among students without attempting to develop benchmarks of such practices.

## Supporting information

**S1 File. Procedures for the collection and analysis of the interview data (qualitative phase of the study) and the generation of initial items for the student behavior questionnaire.** (DOCX)

**S2 File. The student behavior questionnaire in both English and Russian.** (DOCX)

**S3 File. Data set.** (SAV)

**S1 Table. Descriptive statistics of the sample.** (XLSX)

**S2 Table. Results of CFA (factor loadings).**
(XLSX)

**S3 Table. Correlation matrix for the 54 items of the questionnaire.**
(XLSX)

**S4 Table. Covariance matrix for student behavior dimensions.**
(XLSX)

## Author Contributions

**Conceptualization:** Natalia Maloshonok.

**Formal analysis:** Natalia Maloshonok.

**Methodology:** Natalia Maloshonok.

**Project administration:** Kseniia Vilkova.

**Writing – original draft:** Natalia Maloshonok.

**Writing – review & editing:** Natalia Maloshonok, Kseniia Vilkova.

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
