## [Decision Letter · Decision Letter 0]

6 Sep 2024

PONE-D-24-23538Student behavior at university: the development and validation of a 10-dimensional scalePLOS ONE

Dear Dr. Maloshonok,

Thank you for submitting your manuscript to PLOS ONE. After careful consideration, we feel that it has merit but does not fully meet PLOS ONE’s publication criteria as it currently stands. Therefore, we invite you to submit a revised version of the manuscript that addresses the points raised during the review process.

**ACADEMIC EDITOR: **

Dear Authors

The article was reviewed by 2 different reviewers and the opinions are not uniform.

I would like to pay attention to reviewer 1's comments, they are detailed and refer to the length of the article and its various sections. Pay special attention to the comments regarding the findings.

We will wait for the updated version in resubmission.

We look forward to receiving your revised manuscript.

Kind regards,

Gal Harpaz, Ph.D.

Academic Editor

PLOS ONE

Journal Requirements:

"the Basic Research Program of the National Research University Higher School of Economics (HSE University)"

"Support from the Basic Research Program of the National Research University Higher School of Economics (HSE University) is gratefully acknowledged."

"the Basic Research Program of the National Research University Higher School of Economics (HSE University)"

Reviewers' comments:

Reviewer's Responses to Questions

**Comments to the Author**

1. Is the manuscript technically sound, and do the data support the conclusions?

Reviewer #1: Partly

Reviewer #2: Yes

2. Has the statistical analysis been performed appropriately and rigorously? 

Reviewer #1: I Don't Know

Reviewer #2: Yes

3. Have the authors made all data underlying the findings in their manuscript fully available?

Reviewer #1: Yes

Reviewer #2: Yes

4. Is the manuscript presented in an intelligible fashion and written in standard English?

Reviewer #1: No

Reviewer #2: Yes

5. Review Comments to the Author

Reviewer #1: This paper that presents the results of a study undertaken in highly selective universities in Russia is based on the crucial role of students’ university experience in developing their knowledge, skills, and personal and professional growth. My reading of the paper is that as such, it investigates the role of student behaviour. The study does this by developing and assessing the validity of a model that integrates environmental and behavioural aspects to measure students’ behaviour. The author/s put forward that a model such as this, that integrates environmental and behavioural aspects, caters for the diversity of students and their university behaviour.

I believe that the presentation of the model would be of value to the academic community, especially in Russia where the study was undertaken. However, before the paper can be published, further large-scale work on it needs to be undertaken. The points below should assist the author/s in so doing.

The paper is very well researched, and the model presented well supported by the literature from which it is drawn. However, the paper is very long in word length, and even more so when the information in the six appendices is included. The paper is quite broad in its coverage, and while well expressed, is deep. The author/s need to refine the paper including the appendices to streamline it and render it more accessible to readers.

There are some discrepancies in the paper that require review. For instance, it is stated earlier in the paper that eight universities participated in the study, but in Section 3.2 this number is stated as seven. The concept of ‘highly selective’ universities in Russia needs definition as it may differ across the countries of the journal’s readership. A brief explanation of the notion of ‘convenience sampling’ would also be useful for readers.

The paper needs to be reviewed carefully so that there is not repetition of information across it. For instance, some information in the Methods section has been presented earlier in the paper.

The abstract mentions self determination theory, but this does not seem to be discussed in the paper.

Further development of the diagram in Figure 1 would enhance the paper, so that the diagram includes the research areas from which the five behavioural indicators emanate.

The results section should include comment on the value of the sections of the model in relation to the purpose of the paper, which is to examine its validity in measuring students’ behaviour.

The conclusion in the paper is brief and should be expanded to discuss the aim of the model developed as presented on page 4 (line 64), which is to improve students’ development.

I am unclear if the end note referencing system used in the paper is the style required by the journal. The validation of the statistical analysis undertaken in the study has also not been undertaken in this review.

In all, the paper is well researched, and the model developed logical and well supported. The paper is well structured and clearly expressed. However, my reading is that it is a complex and overly long paper which requires further work to refine and streamline it. I wish the author/s well as they proceed further with the paper.

Reviewer #2: The authors Maloshonok & Vilkova of the manuscript titled "Student behavior at university: the development and validation of a 10 dimensional scale" provided a complex study analyzing the student behavior in Russian Universities by using large data collection. I strongly agree with this study in the sense that the combination of environmental and behavior factors are the keys that will shape the students growth and quality. For the least it is a significant contributor to this endeavor. I find this study solid. Authors provided a clear flow of ideas from the introduction until the conclusion. The statistical methodology is strongly supportive of the given hypothesis - assumptions in the introduction.

I have provided some review tracks & changes from the original MS word manuscript to help authors in the English language standard. There are only few questions in comments section from the review and some recommendation.

The study is complete and comprehensive enough beyond expectation. Congratulations.

Please see the attached document and i hope that it does help a little to enhance this valuable manuscript done from a hard working researcher team.

Thank you.

6. PLOS authors have the option to publish the peer review history of their article (what does this mean?). If published, this will include your full peer review and any attached files.

Reviewer #1: **Yes: **Dr Glenda Crosling

Reviewer #2: **Yes: **EXELIS MOISE PIERRE

---

## [Author Response · Author response to Decision Letter 0]

18 Oct 2024

Dear Dr. Gal Harpaz, Dr Glenda Crosling, and Exelis Moise Pierre,

Thank you so much for your thorough reading and helpful comments on this manuscript. We greatly appreciate your help in improving the quality of the manuscript! The detailed answers to your comments and description of changes in the text are presented in the file "Response to Reviewers" in the submission.

Best regards, Natalia Maloshonok

---

## [Editor Report · Decision Letter 1]

23 Oct 2024

Student behavior at university: The development and validation of a 10-dimensional scale

PONE-D-24-23538R1

Dear Dr. Maloshonok,

We’re pleased to inform you that your manuscript has been judged scientifically suitable for publication and will be formally accepted for publication once it meets all outstanding technical requirements.

Kind regards,

Gal Harpaz, Ph.D.

Academic Editor

PLOS ONE

Additional Editor Comments (optional):

Dear Authors,

Thank you for your thorough and professional consideration of the reviewers' comments.

After examining the revised version of the article, and the reference to the reviewers' comments in the body of the revised article itself, I decided to accept the article for publication in its current version.
---

## [Editor Report · Acceptance letter]

6 Nov 2024

PONE-D-24-23538R1 

PLOS ONE

Dear Dr. Maloshonok, 

I'm pleased to inform you that your manuscript has been deemed suitable for publication in PLOS ONE. Congratulations! Your manuscript is now being handed over to our production team.

Kind regards, 

on behalf of

Dr. Gal Harpaz 

Academic Editor

PLOS ONE